# Generalized Approach towards Secretion-Based Protein Production via Neutralization of Secretion-Preventing Cationic Substrate Residues

**DOI:** 10.3390/ijms23126700

**Published:** 2022-06-15

**Authors:** Hyunjong Byun, Jiyeon Park, Benedict U. Fabia, Joshua Bingwa, Mihn Hieu Nguyen, Haeshin Lee, Jung Hoon Ahn

**Affiliations:** 1Department of Biological Sciences, Korea Advanced Institute of Science and Technology (KAIST), Daejeon 34141, Korea; hyunjongbyun@kaist.ac.kr; 2Department of Chemistry and Biology, Korea Science Academy of Korea Advanced Institute of Science and Technology, Busan 47162, Korea; jyp131@kaist.ac.kr (J.P.); fabiabenedict@gmail.com (B.U.F.); jbmakhanu10@gmail.com (J.B.); hieuthao1203@gmail.com (M.H.N.); 3Department of Chemistry, Korea Advanced Institute of Science and Technology (KAIST), Daejeon 34141, Korea

**Keywords:** ABC transporter, type I secretion system (T1SS), protein secretion, recombinant protein expression, protein mutagenesis

## Abstract

Many heterologous proteins can be secreted by bacterial ATP-binding cassette (ABC) transporters, provided that they are fused with the C-terminal signal sequence, but some proteins are not secretable even though they carry the right signal sequence. The invention of a method to secrete these non-secretable proteins would be valuable both for understanding the secretory physiology of ABC transporters and for industrial applications. Herein, we postulate that cationic “supercharged” regions within the target substrate protein block the secretion by ABC transporters. We also suggest that the secretion of such substrate proteins can be rescued by neutralizing those cationic supercharged regions via structure-preserving point mutageneses. Surface-protruding, non-structural cationic amino acids within the cationic supercharged regions were replaced by anionic or neutral hydrophilic amino acids, reducing the cationic charge density. The examples of rescued secretions we provide include the spike protein of SARS-CoV-2, glutathione-S-transferase, streptavidin, lipase, tyrosinase, cutinase, growth factors, etc. In summary, our study provides a method to predict the secretability and a tool to rescue the secretion by correcting the secretion-blocking regions, making a significant step in understanding the physiological properties of ABC transporter-dependent protein secretion and laying the foundation for the development of a secretion-based protein-producing platform.

## 1. Introduction

Most laboratories in academia and biotechnology companies utilize non-secretory *Escherichia coli* systems for protein preparations [1]. In such methods, *E. coli* transformed with an expression vector is cultured and lysed, and during this lysis a protein of interest is unavoidably mixed with other contaminant proteins (see Figure 1a). Thus, in these conventional systems, a target protein can only be prepared via multi-steps of column chromatography, salting out, and dialysis [2]. Broad ranges of proteins are prepared by this *E. coli* strategy, with examples including most laboratory enzymes (e.g., lipases), cell-signaling proteins (e.g., fibroblast growth factor, bone morphogenic proteins), fluorescent proteins (e.g., green fluorescent protein), tag proteins (e.g., streptavidin and thrombin), and countless others.

The *E. coli* protein preparation system almost guarantees to produce functional proteins of interest, unless they require a large degree of post-translational modifications. This preparation fidelity is the primary reason for widespread usages of *E. coli* systems in most laboratories [3]. However, multiple, labor-intensive, and time-consuming purification steps accompanying cell lysis is a main disadvantage.

One solution to overcome this long-time, un-addressed challenging task is secretion of a target protein into extracellular space (see Figure 1b). In these bacterial secretion-based production schemes, the host microorganism secretes the target protein to the medium so that the other cellular proteins remain in the interior of the cell. In this case, intensive purification steps are no longer necessary, greatly saving time, materials, and human resources [4,5]. *Pseudomonas fluorescens* is a good microorganism for this aim [3]. The host is a generally non-pathogenic bacterium [6] with a good tolerance to high-density cell culture [7,8]. It carries an ATP-binding cassette (ABC) transporter [9], which is a type I secretion system (T1SS) capable of secreting many heterologous proteins, such as epidermal growth factor [10], green fluorescent protein (GFP) [11], alkaline phosphatase [12], phospholipase A1 [13], and maltose-binding protein [14].

However, current bacterial secretion systems, including the ABC transporters, are protein-dependent, which means that not all proteins are able to be secreted even though the right signal is attached [5]. Due to this reason, most researchers (even at this time) still use the reliable *E. coli* system [3]. If the criteria separating the secretable and non-secretable cargo protein are identified, then researchers and industry workers can predict whether a specific cargo protein can be prepared via secretion-based methods, even before experimentally testing it. This would greatly reduce the time and personnel costs.

During brief examinations, we noticed that the non-secretable proteins almost always had high-density cationic supercharged regions, or “cationic supercharged regions”. We suspected that this might be the cause of the non-secretability of these proteins, as illustrated in Figure 1c. We assumed that we could identify those cationic supercharged regions by examining the charge density distribution of a linearized polypeptide (see Figure 1d,e), and drawing them as graphs. With these informative graphs (such as Figure 1f,g), we noticed that all non-secretable proteins indeed contained at least one cationic supercharged region.

Herein, we present a step forward to developing a reliable secretion-based protein production method, by suggesting a method to predict whether a secretion-based production is possible before actual experiments, and by providing tools to rescue the non-secretable proteins. We experimentally demonstrate the secretion-blocking property of cationic supercharged regions by adding artificial cationic supercharged regions to secretable proteins. We also identify the secretion-impeding cationic supercharged regions within a few known non-secretable proteins. In detail, we scan for these cationic supercharged regions by examining local charge densities of a linearized amino acid sequence of a target protein, or “linear charge density (LCD)”. In this LCD analysis, a cationic supercharged window is defined as when summation of overall charges in a 20-amino acid box (a “window”) exceeds +2. We could repeat this calculation for all possible windows throughout the entire amino acid sequence of a protein (see Figure 1e again). This process is automated by a computer code created by us. We demonstrate that regions where the LCD values are higher than +2 (i.e., cationic supercharged regions) inhibit the secretion of a protein, exemplifying this in several proteins, including the SARS-CoV-2 receptor binding domain. After the identification of cationic supercharged regions, the corresponding cationic amino acids are mutated into negatively charged or neutral amino acids to reduce the windows’ overall charge to ≤+2. Finally, we demonstrate secretion-enhancing strategies by rescuing secretions of non-secretable, yet valuable proteins, such as glutathione S-transferase, streptavidin, cutinase, MelC2 tyrosinase, M37 lipase, TNFβ, and the spike protein of SARS-CoV-2, etc.

## 2. Results

### 2.1. Conversion of SARS-CoV-2 Spike Protein into Secretable Variants

The spike protein of SARS-CoV-2 is a vaccine candidate, which underlines the importance of its preparation considering the current COVID-19 pandemic and potential future coronavirus outbreaks. As previously shown in Figure 1g, nine cationic supercharged regions which can potentially inhibit its secretion were identified by LCD analysis. Two cationic supercharged regions were identified in both the N-terminal domain (NTD) and the receptor binding domain (RBD) (see Figure 2a). Due to the presence of the supercharged regions (see Figure 2d,e, top row), the unmutated proteins of NTD and RBD were not secreted (see Figure 2f,g, 1st and 4th lanes). Thus, we designed mutants of these proteins to eliminate the aforementioned cationic supercharged regions, by replacing cationic lysine or arginine into anionic aspartate or glutamate. In this way, overall charges of each window (20 amino acid) were suppressed below +2 (see Figure 2b,c, green lines) or +1 (see Figure 2b,c, red lines). Western blot results clearly showed that each of the ≤+2 mutants of NTD and RBD were secreted into extracellular medium, and the secretions were further enhanced in the ≤+1 mutants (see Figure 2f,g).

### 2.2. Introducing Artificial “Cationic Supercharged Regions” to Substrate Proteins

The aforementioned hypothesis, supported by a clear representative demonstration using SARS-CoV-2 spike protein, suggested that the cationic supercharge (over +2) can be a determinant factor for protein secretion with the ABC transporter. If the hypothesis were to be true, insertion of oligopeptide sequences consisting of cationic amino acids in the middle of a well-secreted substrate protein would block its secretion. To test this, we added the following four oligopeptides at C-terminal sides of the cargo proteins, just upstream of the LARD3 signal sequence: Arg05 (five consecutive arginine residues), Arg10, Lys05, and Lys10 (see Figure 3a–e for the structures). We used green fluorescent protein (GFP) as a model substrate. In Figure 3f,g, note that GFP was well-secreted before the insertion of these amino acid patches. However, after the attachment of Arg05, Arg10, and Lys10 sequences, their secretions stopped. The linear charge density “peaks” that these oligopeptides introduced are illustrated in Appendix B Figure A1a–e. Meanwhile, Lys05 sequence seemed to have little effect on GFP secretion. This suggests that lysine exhibits a much weaker secretion-blocking effect than that of arginine. In addition, we performed identical experiments with two different model cargo proteins with better secretion than GFP, namely the *P. fluorescens* ABC transporter’s native substrate TliA or the negatively charged mutant of GFP named GFP(−30) [14]. The results were similar to that of GFP: Arg05, Arg10, and Lys10 blocked the secretion, while Lys05 had little effect (see Appendix B Figure A2a–d).

### 2.3. Rescuing Secretion by Adding Negative Charges near the Positive Charges

We immediately questioned if the secretion-blocking effect of the arginine and lysine oligopeptides could be rescued by proximal anionic charges. We attached the following four sequences to the C-terminus of the model protein GFP: ArgAsp10 (ten repeats of arginine–aspartic acid pairs), ArgAsn10 (ten repeats of arginine–asparagine pairs), LysAsp10 (ten repeats of lysine–aspartic acid pairs), and LysAsn10 (ten repeats of lysine–asparagine pairs) (see Figure 3h–l for the structures, and Appendix B Figure A1f–i for the LCD graphs). Indeed, ArgAsn10 and LysAsn10 blocked the secretion, just like the Arg10 and Lys10 sequences tested in Figure 3a. On the other hand, both ArgAsp10 and LysAsp10 did not harm the secretion, indicating that the adjacent aspartic acid residues were able to neutralize the secretion-blocking effect of the arginine and lysine residues. Furthermore, we performed identical experiments with two additional substrate proteins, TliA and GFP(−30). Both of these additional models yielded identical results (see Appendix B Figure A2e–h).

### 2.4. “Negatively Supercharging” Poorly Secreted Proteins

So far, we have demonstrated that the existence of a cationic supercharged region within the protein sequence blocks the secretion, which was rescued by adding negative charges nearby. We tested if this secretion-rescuing strategy can be generally applied for other various substrates. Our first plan was to mutate the target proteins by providing a significant amount of negative charges, because there is already an existing algorithm to perform this mass addition of negatively charged residues [15]. Using this method, surface-exposed cationic residues were selected as target residues for replacement with anionic amino acids. We first demonstrated that this strategy indeed works for glutathione S-transferase (GST) and streptavidin (SAv), which are intrinsically non-secretable (see Figure 4a, black lanes) due to the existence of cationic supercharged regions. Conveniently, “negatively supercharged” mutant proteins of these two were already previously reported as GST(−20) and SAv(−10) [15]. Thus, we directly proceeded to compare the secretion of the wild-types and these pre-made variants. Indeed, the negatively supercharged mutants exhibited superior secretion compared to the native ones (see Figure 4a, red lanes). The surface charges reversed by negative supercharging in the previous report lowered the peaks in the LCD graphs of GST and SAv (see Appendix B Figure A3c,d).

Using the same algorithm used to create GST(−20) and SAv(−10), we were able to create a negatively supercharged variant of *Nectria haematococca* cutinase (Cuti), *Photobacterium lipolyticum* M37 lipase (M37), and *Streptomyces antibioticus* MelC2 tyrosinase (MelC2). The method was to replace solvent-exposed cationic residues into anionic amino acids, as exemplified in Figure 4b. The results were promising: even though the native Cuti, M37, and MelC2 were poorly secreted, the mutated Cuti(−), M37(−), and MelC2(−) became well-secreted (see Figure 4c,d). Notably, the mutant Cuti(−) retained its intrinsic enzymatic activity, and with the increased secretion, Cuti(−) culture’s extracellular cutinase activity increased (see Figure 4e,f). The LCD graph analysis confirmed that the charge-flooding supercharging mutations completely removed the cationic supercharged regions from cutinase, MelC2 tyrosinase, and M37 lipase (see Appendix B Figure A3e–g). Overall, these results indicated that it is possible to rescue the secretion by flooding the target protein with anionic amino acids, i.e., negatively supercharging them.

### 2.5. “Super-Neutralization”: An Alternative Secretion-Rescuing with Less Structural Disturbance

Although the negative supercharging approach for target proteins made them secretable, reversals of the surface cationic charges into anionic charges in multiple residues can potentially cause elimination of intrinsic protein functions. The intrinsic enzymatic activity was retained for the previous case of cutinase (see Figure 4e), but M37 lipase lost the enzymatic activity by negative supercharging (as detailed below). Supercharging processes often involve a reversal of the charge from cationic to anionic, which can lead to a loss of protein function. In contrast, changing cationic residues into neutral hydrophilic ones would likely cause a decreased level of structural perturbation. To implement this idea, we replaced solvent-exposed cationic residues of M37 lipase into neutral hydrophilic glutamine (single letter code, Q). This mutant was named M37(Q). Likewise, MelC2(Q) was prepared from *S. antibioticus* MelC2 tyrosinase. We termed this approach “super-neutralization”. The result showed successful secretion of M37(Q) and MelC2(Q) (see Figure 5a,b, orange lanes), and importantly, the mutated M37(Q) retained the enzymatic activity (see Figure 5c). The LCD analyses showed that the super-neutralization processes removed the cationic supercharged regions in both M37(Q) (see Figure 5d) and MelC2(Q) (see Figure 5e). The super-neutralized mutants demonstrated that replacing cationic residues with the neutral hydrophilic amino acid, glutamine (Q), can also rescue the secretion and, in addition, has less effects on enzymatic activity than the negative supercharging approach.

### 2.6. Random Mutation and Activity Screening: Avoiding Structure-Disrupting Mutations

From the above experiments, we verified that mutating all the solvent-exposed cationic residues can rescue the secretion of non-secretable proteins. However, this approach unavoidably results in excessive, unnecessary mutations of a given poorly secreted protein. We postulated that we can selectively collect secretion-promoting mutations by ruling out any activity-disrupting mutations. This approach might be similar to a screening process in a directed evolution. We used the mixed-based gene synthesis technique to insert either lysine or glutamate randomly at the solvent-exposed lysine residues. Then, we screened for the clone that exhibited the most prominent halo in the lipase activity assay. The resulting clone, M37(var)—“var” from “variable mutation”, was then characterized via DNA sequencing. M37(var) had both excellent secretion (see Figure 6a, orange lanes) and retained enzymatic activity (see Figure 6b). In fact, the culture supernatant enzymatic activity in Figure 6b actually increased compared to the wild-type, possibly due to the increased amount of enzyme secreted to the culture supernatant. The LCD graph of the M37(var) protein is presented in Appendix B Figure A3h. The SDS-PAGE gel staining experiment in Figure 6c shows that the culture supernatant of the M37(var) mutant mostly consists of M37 lipase enzyme, clearly demonstrating the concept of relatively contaminant protein-free supernatant in secretion-based protein production.

### 2.7. Linear Charge Density-Focused Rational Mutant Design

All the above experiments targeted highly solvent-exposed residues in the protein, but not necessarily all those residues were in cationic supercharged regions. This means that the above methods might involve unnecessary mutations at positions that were already secreted well. Applying the novel charge density hypothesis, we aimed to pin-point cationic supercharged regions within the target proteins and only modify the solvent-exposed cationic residues within those regions. This way, any unnecessary mutations could be avoided. As can be seen from our proof-of-concept case of SARS-CoV-2 spike protein (see Figure 2), we arbitrarily defined cationic supercharged regions as peaks in the LCD graph with height above +2. Relatively smaller-sized, non-secretable *Homo sapiens* signaling proteins, namely transforming growth factor beta (TGFβ), tumor necrosis factor beta (TNFβ), and fibroblast growth factor I (FGF1), were chosen as model proteins. The LCD graph before and after the mutations revealed that we precisely removed the cationic supercharged regions and left the rest of the protein unmodified (see Figure 7a–c). The secretion patterns of the mutants showed that all of the supercharged mutants, TGFβ(−), TNFβ(−), and FGF1(−), were secreted well (see Figure 7d,e, red lanes), while the native versions were not (see Figure 7d,e, black lanes).

### 2.8. Other ABC Transporters Exhibited Similar Behaviors

To verify that the substrate charge density dependency was not limited to the *P. fluorescens* ABC transporter, we tested the secretion of negatively supercharged cargo proteins through a few different polypeptide-secreting ABC transporters. The complexes we tested were as follows: *Pseudomonas aeruginosa* AprDEF, *Dickeya dadantii* (also known as *Erwinia chrysanthemi*) PrtDEF, and *E. coli* HlyBD + TolC. We had already verified that the LARD3 signal sequence we used in this experiment was compatible with these transporters (see Appendix B Figure A4). The sequence identities of these ABC transporters to *Pf*TliD (the one we used in the other experiments of this study) were as follows: *Pa*AprD 57.47%, *Dd*PrtD 55.77%, and *Ec*HlyB 23.64%. We chose the cutinase (Cuti) as the model protein because we could easily detect its secretion in the plate activity assay. We prepared double-transformed *E. coli* cells harboring the genes for the target proteins as well as the ABC transporter genes. Colonies of these cells were tested for secretion via the plate enzymatic activity assay (see Figure 8a,b) and Western blot (see Figure 8c,d). We also tested the supercharged protein M37(−) and wild-type M37 in Western blot and showed similar results (data not shown). The results showed that negatively supercharging the cargo proteins also improved the secretion in all three of the other bacterial ABC transporters.

## 3. Discussion

In this study, we aimed to identify the exact conditions of heterologous cargo proteins that could be secreted by bacterial ABC transporters. We believe that our results disclosed that the “absence of a cationic supercharged region” is the determinant factor that converts poorly secreted proteins to well-secreted ones. Our theory was able to explain all results from previous studies as well. For example, eliminating cationic supercharged regions along the amino acid sequences results in shifting the proteins’ overall charges down to anionic states, which might explain why previously known natural substrates of ABC transporters (which are secreted) tend to be negatively charged [16].

Previous statements to predict the secretability of a cargo protein have been the following:

**Hypothesis** **1** **(H1).***Proteins that are highly negatively charged overall are secreted, otherwise, the secretion is not allowed [14,16]*.

However, our study proposes the following revised general statement that can predict secretion properties of a given protein:

**Hypothesis** **2** **(H2).**
*Any region with a high cationic local charge density within a cargo protein inhibits the secretion. Otherwise, the secretion is allowed.*


According to Hypothesis 1, CTP-TliA and NKC-TliA were categorized as secretable proteins, but they were in fact non-secretable ones [14]. However, our new Hypothesis 2 can properly categorize CTP-TliA and NKC-TliA as non-secretable proteins as both of these proteins carry cationic supercharged regions (see Figure 1f). The novel “absence of a cationic supercharged region” hypothesis is also consistent with the fact that non-secretable proteins cannot be rescued by simply fusing highly anionic proteins at the polypeptide termini (see Appendix B Figure A5). Although these fusions might add some more anionic charges overall, they would not necessarily suppress the cationic supercharged regions within the proteins of interest.

We had briefly discussed the presumptive role of the membrane potential in our previous publication [14]. Back then, we had too little evidence to explicitly state that the linear charge density of the unfolded cargo protein has a crucial role in the transport. With the evidence in this study, we have enough evidence that the positively supercharged regions within the target protein block the transportation through bacterial ABC transporters. When the cationic supercharged regions of substrates are traversing through the transporter channel [17,18,19], the inside-negative membrane potential applies inward electrostatic force to the polypeptide. When the electrostatic force from the repulsion is larger than the driving force, the polypeptide cannot transverse through the channel.

In Figure 3a, we performed a comparative experiment in which Arg05 clearly blocked GFP secretion, but Lys05 did not. We assume that this is related to the side-chain chemistry of these amino acids. In a solvent-exposed state, the conjugate acid p*K*_a_ value of arginine is 12.5 and the conjugate acid p*K*_a_ of lysine is 10.5 [20]. These values are much higher than the physiological cytoplasmic pH of bacteria, which is generally 7.2–7.8 [21,22], including ~7.7 for *Pseudomonas* species [23]. This means that lysine and arginine both exist in protonated states in the cytoplasm. Nonetheless, low-dielectric constant environments, such as inside the phospholipid bilayer membranes or inside proteins, often lower the conjugate acid p*K*_a_ value of basic amino acids [24]. This suggests that the hydrophobic, low-dielectric constant microenvironment inside the transporter channel might allow the deprotonation of lysine, while failing to deprotonate arginine, which have a much higher p*K*_a_.

In fact, other transporters in the Sec transporter family exhibit a similar pattern, where cationic residues are not easily translocated [25,26], and among those cationic residues, arginine is much harder to secrete than lysine [27,28,29]. This could be indirect evidence that the polypeptide-secreting ABC transporters may share a similar fundamental mechanism with the Sec transporters. Both Sec transporters and ABC transporters: (i) involve the configurational changes accompanied by the ATP hydrolysis, (ii) have to “push” the polypeptides through a narrow channel, which allows no complicated structures except for the alpha helices to pass through, and (iii) primarily secrete acidic (or negatively charged) proteins. There was an attempt to explain this behavior as a “proton ratchet” effect in Sec transporters [29], which was an application of the Brownian ratchet mechanism of secretion modes [30,31]. We propose that this theory might also be valid for polypeptide-secreting ABC transporters as well.

As applications of Hypothesis 2, we suggested a few methods to practically “fix” non-secretable cargo proteins. These methods were negative supercharging (see Figure 4), super-neutralization (see Figure 5), random mutation accompanied by activity screening (see Figure 6), and linear charge density-focused supercharging (see Figure 7). They all work by lowering the linear charge density of the target protein via mutagenesis. We demonstrated that all these methods can generate a secretable derivative of a given target protein. The latter three options had a better chance to yield mutants that reserve the original protein’s functionality. Furthermore, many of these secretable variants also showed enhanced overall expression levels (see Figure 3f,m, Figure 5a, Figure 6a and Figure 7e), with few exceptions (see Figure 4a). This phenomenon suggested that removing the product protein from the cytoplasm via secretion might also have an additional benefit in terms of overall expression in some cases, further justifying the development of methods aiming to rescue the protein secretion. Finally, according to the results in Figure 8, the secretion-promoting effect of negatively supercharging cargo proteins was also observed in other polypeptide-secreting ABC transporters. This suggests that the phenomena discussed in this study might potentially be more universal, rather than being specific to the *P. fluorescens* TliDEF ABC transporter complex.

The bacterial ABC transporter’s secretion dependence on the cargo protein’s charge density was studied in various aspects. The results revealed that the absence of cationic supercharged regions in the linear charge density of the cargo proteins, not the overall negative charge, might be the actual determinant of the ABC-dependent secretion. We also provided very potent tools to make proteins secretable by the ABC transporter and thus be compatible with the subsequent secretion-based production. We hope that these results make it possible to predict whether any given protein of interest can be secreted by ABC transporters before actual testing and make it possible to produce it via secretion anyways, even if the prediction results were negative. Together, these discoveries provide a deeper understanding of the characteristics of secretable substrates of bacterial ABC transporters, as well as revolutionize the potential of bacterial ABC transporters as the platform for efficient secretion-based protein production.

## 4. Materials and Methods

### 4.1. Bacterial Strains and Growth Conditions

All major gene works, including the plasmid construction, were performed using the *E. coli* XL1-BLUE strain. Protein expression and secretion were observed in both *E. coli* XL1-BLUE and *P. fluorescens ΔtliA ΔprtA* strains [11]. We performed the *E. coli* transformation with the standard heat shock method, while *P. fluorescens* transformation was performed via electroporation at 2.5 kV, 125 Ω, and 50 μF, with electrocompetent cells prepared using a standard electroporation protocol [32]. M9 minimal media, Lysogeny broth (LB), and terrific broth (TB) supplemented with 30 μg/mL of kanamycin were used for the liquid cultures, which were usually 5 mL in a 180 rpm shaking incubator if not specified otherwise. The liquid cultures were grown until the stationary phase was reached. The protein expression was not regulated by induction. The cells constantly expressed the target proteins during their growth. LB agar plates supplemented with 30 μg/mL of kanamycin were used for the plate colony cultures. In most cases, *E. coli* and *P. fluorescens* were cultured at 37 and 25 °C, respectively, if not specified otherwise. As an exception, *E. coli* enzymatic plate assays (detailed later) were incubated at 25 °C. The cells seeded in liquid culture were incubated until they reached an *A*_600_ of ~3, which is their full growth stage. *E. coli* cells typically take about 1.5 days of growth to reach this stage, while *P. fluorescens* cells require 2 days. The proteins were analyzed for both expression and secretion by seeding the transformed cells in liquid cultures or streaking them on the solid plate activity assay.

### 4.2. Examining the Expression and Secretion of the Proteins

We expressed the prepared plasmids in *P. fluorescens* liquid culture, mainly LB or, less frequently, TB supplemented with 60 μg/mL of kanamycin. The liquid culture was harvested after reaching the stationary phase, which occurs at approximately *A*_600_ = 3–4. The culture was then centrifuged at 18,000 rcf for 10 min, where we took the cell pellet and the liquid supernatant separately, and sample amounts equivalent to 16 μL of original cultures were loaded in SDS-PAGE using 10% polyacrylamide gels. We used nitrocellulose membranes (Amersham, Germany) for Western blotting. Polyclonal anti-LARD3 rabbit immunoglobulin G (rIgG) was utilized as the primary antibody, with 1:3000 dilution in 5% skim milk solution, and anti-rabbit recombinant goat IgG-peroxidase (anti-rIgG goat IgG-peroxidase) was used as the secondary antibody with 1:5000 dilution. The bands were then detected using a chemiluminescence agent (Advansta WesternBright Pico, San Jose, CA). Western blot images were acquired using an Azure C600 automatic detection system (Azure Biosystems, Dublin, CA). All included Western blot images are representative results from at least three different repeated experiments, starting over again from cell culturing with independent colonies. Quantifications of the Western blot images were performed with ImageJ software (https://imagej.nih.gov/ij/, accessed on 11 March 2019), with the unprocessed raw TIFF image data. Within the ImageJ output, the raw integrated density (RawIntDen) of each band was subtracted from a background RawIntDen measured from empty regions with the same area from the identical membrane image. This value was labeled as “calibrated intensity (*I*)”. Then, the % secretion values were calculated as follows:(1)$$\%\;\mathrm{Secretion}=\frac{I_{\mathrm{supernatant}}}{I_{\mathrm{supernatant}}+I_{\mathrm{cell}\;\mathrm{pellet}}}\times 100\%.$$

Statistical analyses were performed with the paired, single-tailed Student’s *t*-test, where the signatures from the same image were paired together. The significance threshold (*α*) was set to be 0.05, which was equivalent to 5% chance of the null hypothesis (no secretion difference between the two compared groups) being true.

### 4.3. Examining the Enzymatic Activities of the Substrate Proteins

We utilized the LB agar plate-based enzymatic activity assay of the expressed and secreted proteins to observe the activity of the recombinant proteins. The colonies of *P. fluorescens* harboring the recombinant plasmids were streaked on the LB agar plate supplemented with enzyme substrates, which will be digested by the enzymes and exhibit visual changes (halo) around the cell colonies. The cells expressing tyrosinases (MelC2 and its derivatives) were streaked on LB agar plates supplemented with 0.1 mM CuCl_2_ and 2 mM l-tyrosine, and the previously mentioned concentrations of appropriate antibiotics. The cells expressing the proteins with triglyceride lipase/esterase activities (M37, cutinase, and their derivatives) were streaked on LB agar plates supplemented with 0.5% colloidal tributyrin (glyceryl tributyrate) and antibiotics. The plates were incubated in a standing incubator at 25 °C for 3 days. The lipase/cutinase activity in the culture supernatant was measured using ρ-nitrophenol-palmitate (pNPP) and ρ-nitrophenol-butylate (pNPB). Then, 5 μL of lipase supernatant was added to 200 μL of pNPP reaction solution and 1 μL of cutinase supernatant to 200 μL of pNPB reaction solution. For the pNPP reaction solution, 10 mM of pNPP in acetonitrile, ethanol, and 50 mM Tris (pH 8.5) were mixed at a ratio of 1:4:95. Similary for the pNPB solution, 10 mM of pNPB in acetonitrile, ethanol, and 50 mM Tris (pH 8.0) were mixed at a ratio of 1:4:95. The reaction solution was incubated for 10 min at 45 °C and monitored at *A*_405nm_. Statistical analyses were performed with Student’s *t*-test, double-tailed and homoscedastic (as the *F*-test failed to deny heteroscedasticity). The significance threshold (α) were set to be 0.05 for both the *t*-test and the *F*-test.

### 4.4. The Positively Charged Patch Experiments

For construction of pBR10 (which adds Arg10), pBK10 (Lys10), pBR05 (Arg05), pBK05 (Lys05), pBRD10 (ArgAsp10), pBKD10 (LysAsp10), pBRN10 (ArgAsn10), and pBKD10 (LysAsp10), we did not use the PCR-based amplification. This is primarily because the sequence we were aiming to insert was short (<30 bp). We first ordered the primers given by the upper half of Appendix A. As these primers overlapped with each other entirely, they created a double-stranded region with the DNA sequence for the insert. The insert region was the desired sequences plus a few extra bases to align the reading frame. In addition, we added a new SacI site in this region, since the original SacI site in the pDART was supposed to be consumed during the In-Fusion cloning step. The inserted region was flanked by single-stranded hangovers, which can complementarily bind to the termini of the SacI-digested pDART plasmid via In-Fusion cloning (Takara, Shiga, Japan). Since these fragments already contain the single-stranded region for binding, we ran the 3′-to-5′ exonuclease reaction step without the insert mix. The insert mix was added immediately after the reaction step was completed. The resulting colonies after the In-Fusion cloning were examined via DNA sequencing, to verify that they were properly constructed. In fact, due to the way the primers complementarily bind, pBKD07 (LysAsp07) and pBKD07 (LysAsp07) were constructed instead of the intended plasmids. Therefore, we repeated the above steps with different primers (also provided in Appendix A) to add three additional repeats to finally construct the intended pBKD10 (LysAsp10) and pBKD10 (LysAsp10). Then, the prepared plasmids were digested with XbaI and SacI restriction enzymes to clone the genes for TliA, GFP, and GFP(−30), using the standard gene cloning method, using the multiple cloning site of the constructed plasmids.

### 4.5. Manual Mutagenesis

We synthesized the DNA sequence of wild-type and supercharged versions of glutathione S-transferase (GST) and streptavidin (SAv), both negatively and positively supercharged, as reported in a previous work of Liu’s group [15]. Additionally, we manually created a less radically supercharged version of positively supercharged GST (GST(+19)) because the version published by Liu’s group had a radical level of supercharging that would be unnecessary for our purpose. The super-neutralization processes (MelC2(Q), M37(Q)) were also performed manually. The structures of the proteins were either downloaded from the Protein Data Bank (database https://www.rcsb.org/, accessed on 8 January 2020) or predicted structures obtained from SWISS-MODEL (https://swissmodel.expasy.org/, accessed on 30 January 2020) [33]. The structures of the proteins were examined, and the charged amino acid residues exposed to the solvent were marked, and these amino acids were converted into positively charged amino acid (for the case of GST(+19)) or glutamine (for the super-neutralized proteins).

### 4.6. Computational Design of Supercharged Proteins

We utilized Liu’s group’s AvNAPSA algorithm [15,34]. We re-wrote the original Perl script into a python 3.9 script and used it to create negatively supercharged versions of *Nectria haematococca* cutinase (Cuti), *Photobacterium lipolyticum* M37 lipase (M37), and *Streptomyces antibioticus* MelC2 tyrosinase (MelC2). AvNAPSA is an abbreviation for “average number of neighboring atoms per side-chain atom” [35]. It was originally developed by Liu’s group to make a protein with resilience to refolding, and the supercharged proteins were also applied for cellular protein targeting in animal cells. In this paper, however, the supercharging protocol was used to generate proteins that are secretable by the ABC transporter complex. We gradually mutated positively charged residues in the increasing order of the AvNAPSA score (low AvNAPSA score means that thre residue has less interaction with other amino acids and that the residue was located on the surface of the protein) until the number of mutated residues was similar to the results we obtained during the manual supercharging process. The exact AvNAPSA thresholds for our mutated proteins are presented in Appendix A. Also note that we excluded any residue proximal (closer than seven residues apart) to the active site residues, cited from their respective literature.

### 4.7. Synthesizing and Cloning the Genes of Supercharged Proteins

We ordered the DNA synthesis service provided by Macrogen (Seoul, South Korea), and Cosmogenetech(Seoul, South Korea). We amplified the genes with PCR and then cloned in the *E. coli*, using the pDART plasmid as the vector. The pDART plasmid was developed in our previous study [12], which acts as a shuttle vector between *E. coli* and *P. fluorescens*. The pDART plasmid also contains the gene for the LARD3 signal sequence, which is conjugated to the C-terminus of the introduced gene, for the recognition by the TliDEF transporter complex. Additionally, the vector enhances the expression level of the TliDEF transporter complex.

### 4.8. Various ABC Transporters Test (Double-Transformation)

We constructed pLARD3 from the commercially available pKK223-3 vector. The multiple cloning site pKK223-3 vector was fused with the *LARD3* gene so that the gene inserted to the multiple cloning site was expressed with a C-terminally fused LARD3 signal sequence. We recombined this pLARD3 with *cuti(−)* and *cuti* genes, prepared previously, to construct pLARD3-Cuti(−) and pLARD3-Cuti. Then, these plasmids were both introduced to *E. coli* along with the plasmids that encode the ABC transporter genes.

### 4.9. Synthesizing and Cloning the Gene of Variationally Supercharged M37

We prepared the DNA sequence of variationally supercharged M37 lipase, which we named M37(var), by replacing the codons for the surface-exposed positively charged amino acid residues with the degenerate codon. For example, the IUPAC DNA code R denotes the purine base, which is guanine or adenine. We aimed to find the Goldilocks variant somewhere between M37(−14), which had enzymatic activity but was not secreted, and M37(−23), which was secreted but had no enzymatic activity. We examined the amino acid sequences of M37(−23) and M37(−14) and marked the residues where the two differed. Then, we replaced codons for those residues with the degenerate codons. For instance, the residue Lys36 of M37 lipase was replaced by glutamic acid in M37(−23) but remained unchanged in M37(−14). Therefore, we placed the degenerate codon “RAG” at that position. For the degenerate codon RAG, there were two possible outcomes: GAG, which codes for glutamic acid, and AAG, which codes for lysine. We would have preferred to use the set “glutamic acid or arginine”, but such a combination was impossible. Therefore, we used the RAG codon in place of these residues as well. The exact sequence we ordered for the construction of M37(var) is provided in Appendix A. After the DNA sequence was designed, we used the DNAWorks web server (https://hpcwebapps.cit.nih.gov/dnaworks/, accessed on 18 January 2018) [36] to convert the sequence into a set of synthesizable primers. The parameters we used were as follows: oligo length 58 nucleotides, annealing temperature 62 °C, oligo concentration 1.00 × 10^−7^ M, Na^+^/K^+^ concentration 0.05 M, Mg^2+^ concentration 0.002 M, number of solutions 1, and no “TBIO” mode (“PTDS” mode was used instead). Then, we manually examined the output oligos and made sure that no degenerate codon was present at the end of any overlapping region between oligos. We ordered the oligos from Cosmogenetech, and we assembled them using the PCR-based DNA synthesis method described in a previous publication [37]. The obtained PCR product was purified, restricted, and then introduced into the pDART plasmid like the rest of the genes handled in this study.

### 4.10. Linear Charge Density Analyses

The “linear charge density (LCD)” was defined as the sum of the charges over a given number (*w*, representing “window”) of consecutive amino acids in a single polypeptide chain, divided by *w*. For example, if the window width *w* = 20, then the linear charge density at index *i* (*i* is a positive integer) is the sum of the charges of the 20 amino acids from position *i* to position *i* + 19, divided by 20. We arbitrarily chose *w* = 20. One of the justifications for choosing 20 was that the average alpha helix spanning through a membrane is roughly that long. Thus, we assumed that a linearized polypeptide being translocated through the membrane channel would have at maximum 20 amino acids under the influence of the electric field formed by the membrane potential at a time. The formula for the linear charge density at position *i* (*λ_i_*) was the following:(2)$$\lambda_{i}=\frac{1}{w}\overset{i+w-1}{\underset{j=i}{\sum }}q_{j}\;,$$
where *w* is window width and *q_j_* is the average charge of the side chain of residue *j* at pH 7. We used −1 for aspartic acid and glutamic acid, +1 for arginine and lysine, and +0.1 for histidine. Interestingly, the non-secreted proteins all had “peaks” over +2 (unit was *e*/20 aa, elementary charges per 20 amino acids) at more than one position within their sequence, while almost all the secreted proteins had no such peaks (and as specified in the Discussion Section, these exceptions had their peaks mainly composed of lysines, not arginines). The LCD graphs were obtained using a script that we developed and published as a webserver (https://mb.re.kr/apps/LCD, accessed on 11 May 2022).

### 4.11. The linear Charge Density-Focused Supercharging

Linear charge densities of the target proteins (TGFβ, TNFβ, FGF1, and SARS-CoV-2 spike protein domains) were automatically calculated using a Python script that we created. We identified the peaks with height above +2 *e*/20 aa and aimed to remove those peaks by replacing the non-conserved, positively charged residues within those peaks. The Bayesian conservation score (a measure of the degree of conservation of a given residue among the homologs of the query protein) of each residue was calculated using the ConSurf webserver (http://consurf.tau.ac.il/, accessed on 8 March 2020) [38,39]. Examining the Bayesian conservation scores, the positively charged residues with low conservation scores (<4) around the high linear charge density peaks were selected and replaced. If there was any negatively charged or hydrophilic amino acid that occurred at the position in question among the homologs returned by the ConSurf analysis, we preferred to use that amino acid. Here, we made sure that the residue was solvent-accessible by examining its reported structure in the protein data bank (PDB) database (https://www.rcsb.org/, accessed on 8 January 2020) or predicted structure obtained from SWISS-MODEL (https://swissmodel.expasy.org/, accessed on 30 January 2020) [33]. In addition, we looked through the literature to make sure that the altered residues were not present on the receptor-interacting surface. Finally, we examined the stability of the interaction between the final modified protein and the receptor protein by running the docking molecular dynamics (docking MD) simulation, using the following procedure: The PDB files of the protein of interest were initially preprocessed using the default parameters of the Schrödinger Desmond program (New York, NY) and missing side chains and loops were filled in using Prime. PROPKA pH 7.0 was used to assign the H-bonds of the structure, which were further optimized by the OPLS3e field model. Lastly, the simulation of the ligand-receptor complex was run for 50 ns in Schrödinger Desmond. We published the LCD-focused supercharging tool as a webserver, which can be accessed using the following link: https://mb.re.kr/apps/supercharge (accessed on 11 May 2022).

### 4.12. Protein Structure Prediction

The structures presented in Figure 1 were predicted from the SWISS-MODEL webserver (https://swissmodel.expasy.org/, accessed on 25 November 2018) [33]. In the process, TliD prediction used *Aquifex aeolicus* PrtD (PDB ID 5l22.1) as the template, with sequence identity of 40.98%. TliF prediction used *Salmonella typhi* ST50 (PDB ID 5bun.1) with sequence identity of 26.52%, but with a good QMEAN score of −1.19. LARD3 signal sequence prediction used a *Pseudomonas* species’ MIS38 lipase (PDB ID 2zj7.1) with sequence identity of 53.85%. TliE prediction used *Serratia marcescens* LipC (PDB ID 5nen.1), with sequence identity of 44.19%. In the template 5nen, the membrane fusion protein LipC was crystallized in dimer, which is different from the expected natural homo-hexameric state of the ABC transporter’s membrane fusion proteins, as specified by the authors of 5nen [40]. The structure of the homo-hexamer complex of TliE was predicted using the GalaxyHomomer webserver (http://galaxy.seoklab.org/cgi-bin/submit.cgi?type=HOMOMER, accessed on 12 February 2020) [41], using the SWISS-MODEL predicted structure of monomeric TliE as the input, with the default parameters. We further optimized the result by repeating the optimization service within the GalaxyHomomer server.

## Figures and Tables

**Figure 1 ijms-23-06700-f001:**
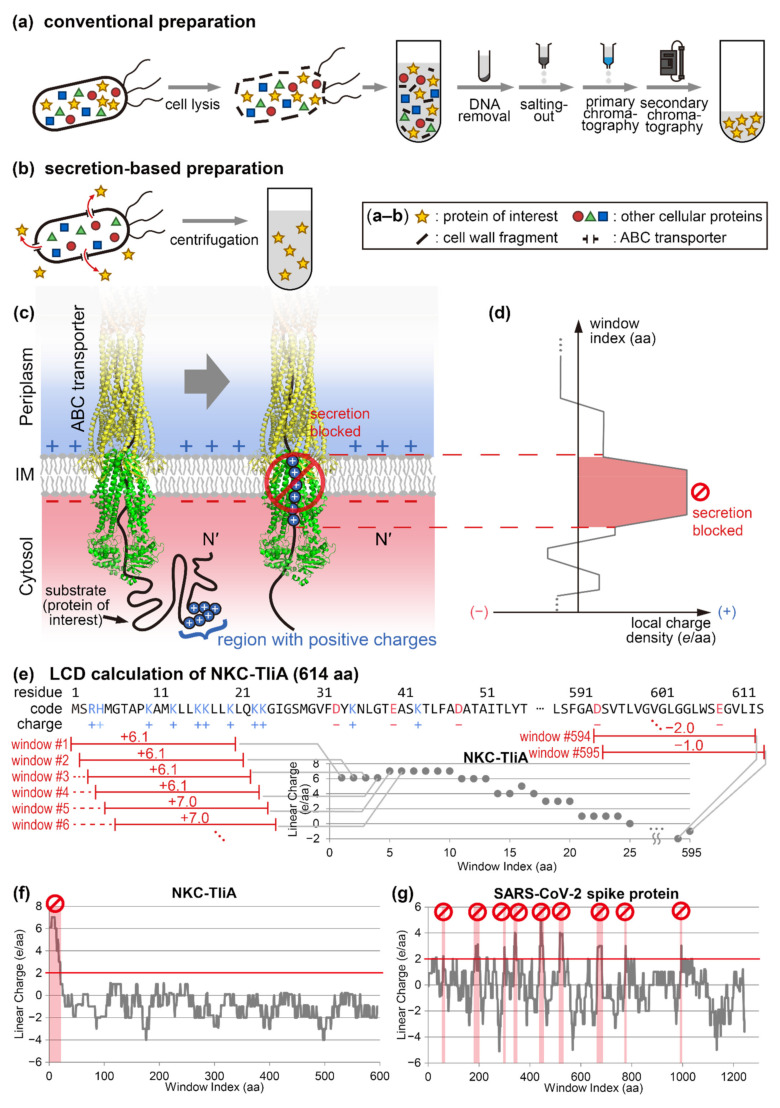
Schematics of this study. (**a**) A summary of conventional protein preparation methods, which involve lysis of the expression host followed by multiple purification steps. (**b**) A summary of secretion-based protein preparation methods. The centrifuged culture supernatant is relatively devoid of contaminants, reducing the complexity and cost of the purification process. (**c**) An illustration of ABC transporter-mediated protein secretion. Some proteins of interest were not compatible with the secretion-based preparation methods, possibly because of the “cationic supercharged” region within their sequences. The inside-negative membrane potential makes it energetically unfavorable to secrete cationic residues. IM: inner membrane. (**d**) The cationic supercharged regions can be identified by analyzing the local charge density of linearized protein, or “linear charge density”. (**e**) Linear charge density (LCD) of a protein at a certain residue (say, residue *i*) is calculated by summing the average charge at pH 7 of the side chains from residue *i* to residue *i* + 19. This block of 20 residues is labeled “window #*i*”. We repeated this from *i* = 1 to *i* = (*n* − 19), where *n* was the length of the polypeptide. (**f**,**g**) LCD graphs of two typical non-secretable proteins, NKC-TliA, CTP-TliA, M37 lipase, and SARS-CoV-2 spike protein. Note that all of them have high peaks, which represent regions crowded with positively charged amino acids (“cationic supercharged” regions).

**Figure 2 ijms-23-06700-f002:**
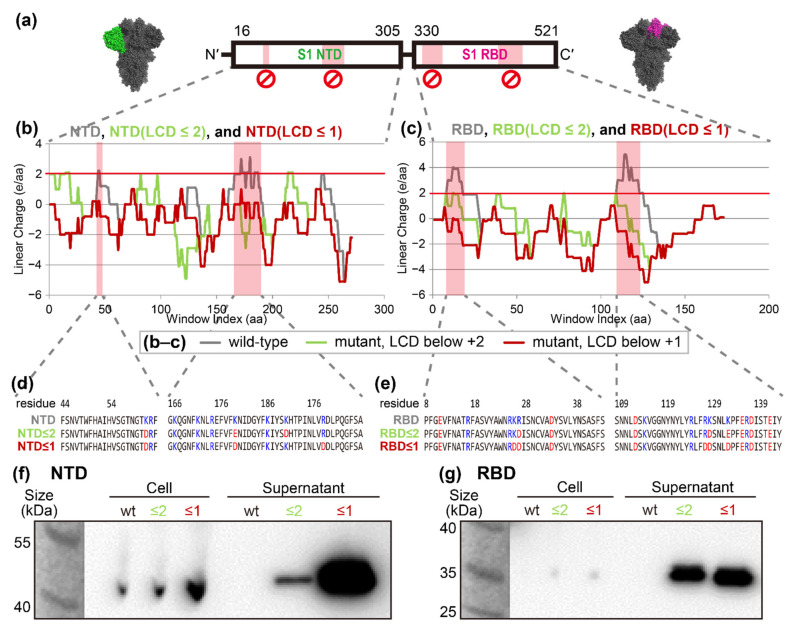
The secretion of SARS-CoV-2 spike proteins and its mutants without cationic supercharged regions. (**a**) The SARS-CoV-2 spike protein homotrimer (PDB ID 6zp0) S1 has two domains: N-terminal domain (NTD, dark green) and receptor binding domain (RBD, pink). (**b**,**c**) Linear charge density (LCD) graphs of native and mutant forms of NTD and RBD (gray lines). The cationic supercharged widows of NTD and RBD are indicated by the red backgrounds. Two mutants, NTD (LCD ≤ 2) and RBD (LCD ≤ 2), were mutated to keep the heights of LCD graphs lower than +2 (green lines). NTD (LCD ≤ 1) and RBD (LCD ≤ 1) were mutated to keep the heights of LCD graphs lower than +1 (red lines). (**d**,**e**) The amino acid sequences of cationic supercharged regions marked in panels (**b**,**c**). (**f**,**g**) The NTD, RBD, and their derivatives were expressed in *P. fluorescens* harboring the ABC transporter. Both cell pellet and supernatant samples were taken and analyzed via Western blot. Darker signal in the supernatant samples implies more secretion.

**Figure 3 ijms-23-06700-f003:**
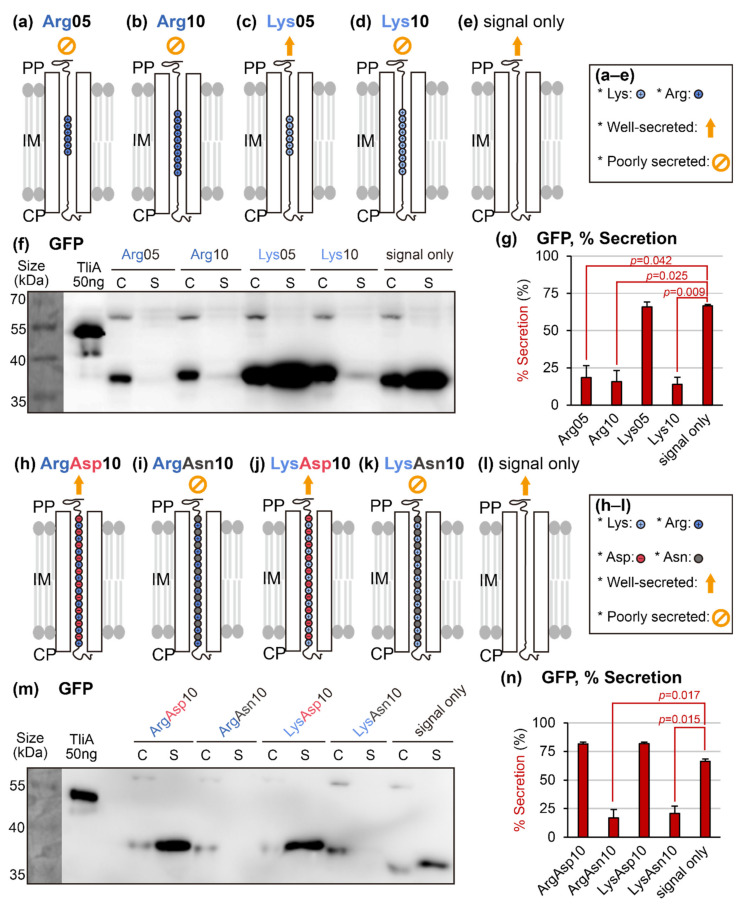
Investigation of secretable conditions by ABC transporters. (**a**–**e**) Structures of the constructed artificial cationic supercharged regions consisting of consecutive cationic amino acids, Arg05, Arg10, Lys05, and Lys10. Yellow arrows indicate that the protein was secreted, while yellow cancellation signs indicate the opposite. (**f**) Western blot analysis of the artificial cationic supercharged regions. The four artificial cationic supercharged regions were inserted into green fluorescent protein (GFP), which is intrinsically secretable. The cell pellet samples are labeled “C”, and the supernatant samples are labeled “S”. Signal in an S sample indicates that the protein was secreted. (**g**) Quantification of experiment presented in panel (**f**). Error bars represent sample standard errors. Performed on biological duplicates, *n* = 3. (**h**–**l**) Structures of artificial peptide sequences used for the “rescue of secretion” experiment, where peptides consisting of alternating cationic–anionic or cationic–neutral amino acids were compared to each other. (**m**) Western blot analysis of the secretion-rescuing experiment using adjacent negative charges. (**n**) Quantification of experiment presented in panel (**m**). Compared to the “signal only” control group. Error bars represent sample standard errors. Performed on biological duplicates, *n* = 3.

**Figure 4 ijms-23-06700-f004:**
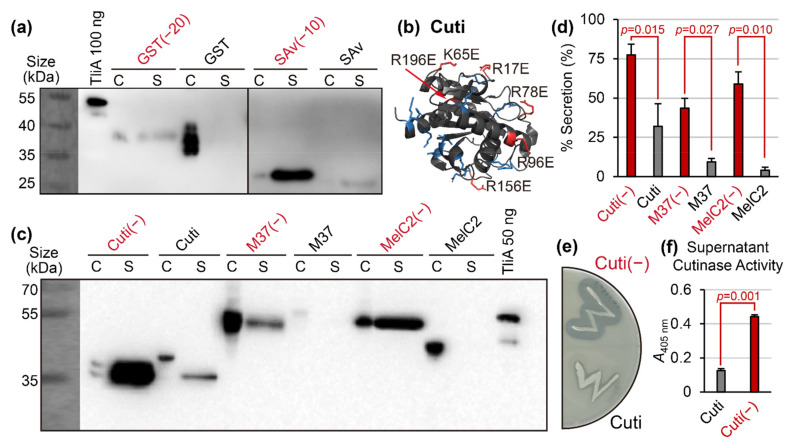
Secretion of negatively supercharged proteins. The liquid cultures of *P. fluorescens* were grown in TB for two days at 25 °C. The cell pellet samples are labeled “C” while the supernatant samples are labeled “S”. A signal in “S” indicates that the sample was secreted. (**a**) Secretion enhancement in negatively supercharged versions of glutathione S-transferase (GST) and streptavidin (SAv). The negatively supercharged mutants were secreted while the wild-types were not. (**b**) The structure of *N. haematococca* cutinase (Cuti) (PDB ID 1cex), used for the supercharging mutation. Red color indicates the surface-exposed cationic residues with AvNAPSA value ≤ 100, which were replaced by anionic glutamates. Blue color indicates the rest of the cationic residues. (**c**) Secretion enhancement by negatively supercharged enzymes. Negatively supercharged mutants of Cuti(−), M37(−), and MelC2(−) were compared with wild-type cutinase, M37 lipase, and MelC2 tyrosinase for secretion using Western blot. Note that Cuti(−) C lane has two bands, probably corresponding to oxidized and reduced forms. (**d**) Quantification of Western blot experiment presented in panel (**c**). All three of the negatively supercharged mutants (red colors) exhibited a significant increase in secretion. Error bars represent sample standard errors. Performed on biological duplicates, *n* = 3. (**e**) Plate enzymatic activity assay of Cuti(−) and Cuti. The negatively supercharged mutant Cuti(−) colony exhibits a much larger halo, due to the increased amount of secreted enzyme. (**f**) Quantification of results in panel (**e**). Cell culture supernatant cutinase activity was measured with the pNPB assay.

**Figure 5 ijms-23-06700-f005:**
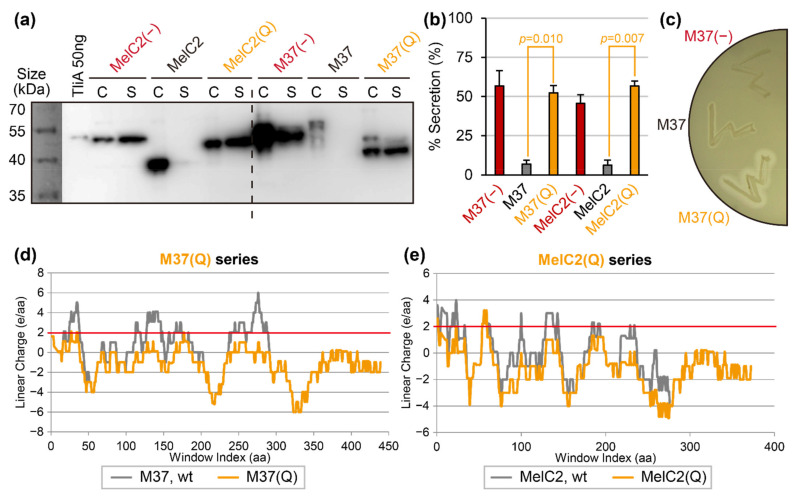
Secretion of proteins which were mildly mutated via super-neutralization. (**a**) The “super-neutralized” mutants, M37(Q) and MelC2(Q), were tested for secretion via Western blot. A signal in “S” implies secretion. (**b**) Quantification of results presented in panel (**a**). Error bars represent sample standard errors. (**c**) Enzymatic activity of M37 lipase and its derivatives developed in this study. A halo indicates that the protein had both been secreted and had enzymatic activity. (**d**) LCD graphs of M37(Q) and native M37 lipase. (**e**) LCD graphs of MelC2(Q) compared to native MelC2.

**Figure 6 ijms-23-06700-f006:**
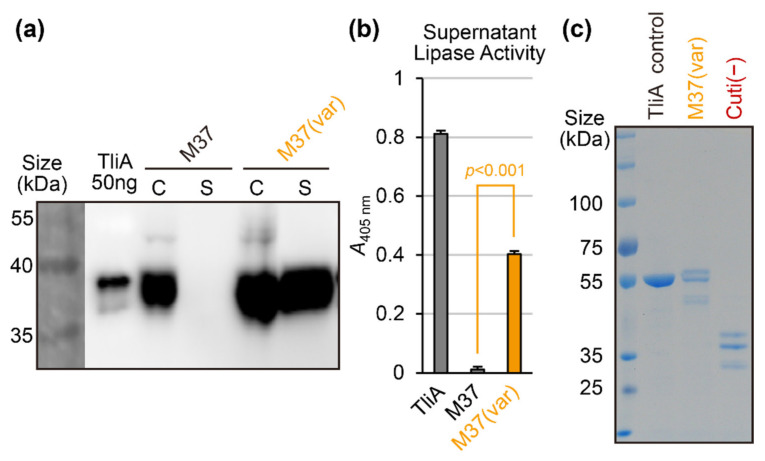
The secretion and supernatant activity of M37(var) lipase and Cuti(−). (**a**) A mutant clone of M37 obtained from random mutation and activity-screening strategy, M37(var), was tested for secretion. (**b**) The activity of the culture supernatant was measured for lipase/cutinase activity of three different enzymes in the supernatant. The activity of TliA and M37 lipase was measured using pNPP and the activity of cutinase using pNPB. (**c**) SDS-PAGE of culture supernatants. 16 μL of culture supernatant was loaded on SDS-PAGE and stained with Coomassie blue. The culture supernatant consisted only of the respective target proteins: M37(var) and Cuti(−). Note that each of these proteins were separated into several bands corresponding to oxidized and reduced forms of LARD3-conjugated proteins (upper two bands) and lower LARD3-cleaved proteins.

**Figure 7 ijms-23-06700-f007:**
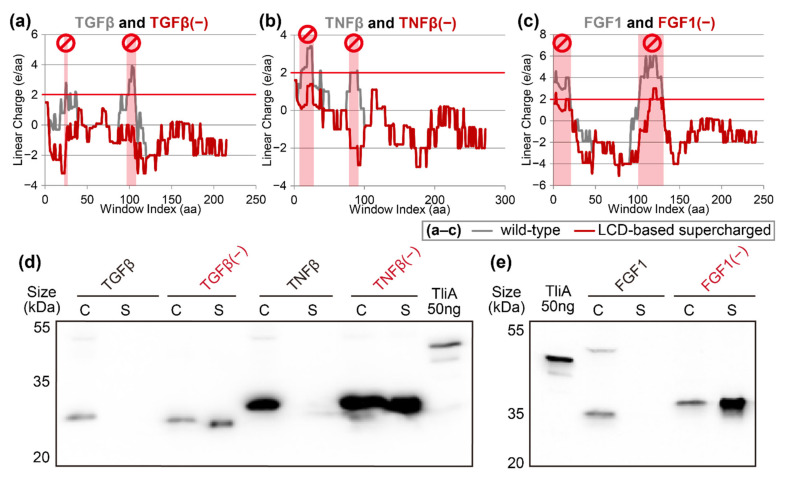
Linear charge density-focused supercharging significantly improves the secretion with minimal mutations. (**a**–**c**) LCD graphs of TGFβ, TNFβ, and FGF1 and their respective mutants. These mutants were designed by identifying >+2 peaks (red background) in the LCD graphs of native proteins (solid gray lines) and mutating positively charged solvent-exposed residues within those peaks. We only mutated residues with low ConSurf Bayesian conservation scores (<4). We also looked upon the homologs of the query protein and saw which amino acids were frequently occurring in that position, to select the replacement among anionic amino acids and neutral hydrophilic amino acids. Highly conserved amino acids (conservation score > 6) remained unmutated in all situations. In addition, the amino acids located in the ligand-receptor binding sites were not mutated. (**d**,**e**) The resulting mutant proteins from panels (**a**–**c**) were introduced into *P. fluorescens* cells and were tested via Western blot.

**Figure 8 ijms-23-06700-f008:**
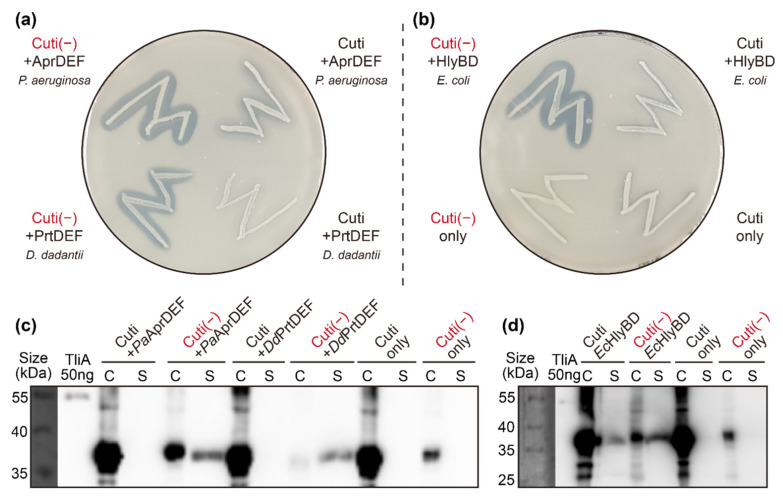
The effect of cargo protein supercharging in the other ABC transporters. *E. coli* XL1-Blue was double-transformed with two plasmids, one expressing cutinase variants and the other expressing the polypeptide-secreting ABC transporter. Empty vector without ABC transporter was used as the negative control. The colonies were screened from LB agar plates containing 50 μg/mL of carbenicillin and 100 μg/mL of chloramphenicol, then streaked on LB agar plates containing 0.5% colloidal tributyrin. (**a**,**b**) The three different ABC transporters, *Pa*AprDEF, *Dd*PrtDEF, and *Ec*HlyBD + TolC complexes, successfully secreted the negatively supercharged cutinase (Cuti(−)). They secreted much less wild-type cutinase (Cuti). This could be clearly seen from the halo size comparisons. (**c**,**d**) The supercharged and wild-type versions of cutinase were tested for secretion via the three different polypeptide-secreting ABC transporters. In these Western blots, both the cell pellet (labeled “C”) and the culture supernatant (labeled “S”) samples were analyzed in parallel.

## Data Availability

The sequences of the constructed mutant polypeptides are provided in FASTA format in Appendix B. The Western blot raw data are provided in Appendix A.

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
