# Peer review of "Generalized Approach towards Secretion-Based Protein Production via Neutralization of Secretion-Preventing Cationic Substrate Residues"

_ijms, 2022, doi:10.3390/ijms23126700_

Round 1

Reviewer 1 Report

The study is a well designed, systematic analysis of the role of linear charge distribution in proteins when secretion is required in the applied bacterial secretion type expression system. I have only a few minor comments:

1. On Fig 4c the labeling is shifted, should be corrected. I think also, that more explenation is needed for this experiment. The presented western blot shows that Mwt of the secreted form is different from the "in cell" fraction - probably the signal sequence is cutted outside, but it is not mentioned. M37 protein shows a very weak expression - it seems that probably not only the secretion but expression is better for M37(-), some explanation is required. (Similar phenomenon can be seen with other proteins)

2. On Fig6A - it seems that marker labeling is fault.

3. On Fig 8 it seems that C samples contains very different amount of proteins - again expression level of the protein may alter the secretion of protein as well. In this experiment ABC transporters are expressed from plasmids as well but proof of ABC transporter expression is not presented. Moreover a control would be interesting without ABC transporter overexpression as supplemental data.

Author Response

Thank you for the valuable critiques you provided. We hereby send you a point-by-point response to your review. Your original comment is colored black, while our response is colored red.

  1. On Fig 4c the labeling is shifted, should be corrected. I think also, that more explenation is needed for this experiment. The presented western blot shows that Mwt of the secreted form is different from the "in cell" fraction - probably the signal sequence is cutted outside, but it is not mentioned. M37 protein shows a very weak expression - it seems that probably not only the secretion but expression is better for M37(-), some explanation is required. (Similar phenomenon can be seen with other proteins)

The labeling of Figure 4c was adjusted to match the lanes, as requested.

The apparent difference in sizes of supernatant (extracelluar) and intracellular proteins could be from the secretion signal peptide cleavage, but we think this is not likely the case. The bacterial type I secretion systems does not normally cleave the signal sequence, and our western blot antibody was targeted against the signal sequence. Our best speculation is that the reduced and oxidized forms of same polypeptide co-exist, having different travel speeds in SDS-PAGE. As can be seen from Fig 6c, Fig 5a, Fig 4a, and especially Fig 4c Cuti(−) Cell sample, some proteins have double bands, both of which appear in western blots. The caption of Fig 4c (line 230) was modified to accommodate this. Another supporting evidence is that proteins lacking cysteine residues (e.g. MelC2 tyrosinase and its variants) do not have this multiple band issue.

It was a very good point that there seem to be an expression level difference between poorly secreted and well-secreted proteins, which seems to be our next big topic for a future study. Indeed, this expression level difference was not limited to this image, as can be seen in other images like Fig 3f, Fig3m, Fig 5a, Fig 6a, Fig7e, and Supplementary Text S2 (raw data collection) p. 25. We believe that there could be two reasons behind this phenomenon. Firstly, the secreted proteins are relatively protected from the intracellular proteases, resulting in less turnover and thus elevated overall expression. Secondly, some of the proteins (e.g. M37 lipase) could be slightly toxic to the cell, which may cause damage to the cell when accumulated in the cytoplasm. We agreed that this effect was worth mentioning, and we added a couple of sentences summarizing this effect in the main text, at line 416.

  1. On Fig6A - it seems that marker labeling is fault.

The marker labeling was corrected.

  1. On Fig 8 it seems that C samples contains very different amount of proteins - again expression level of the protein may alter the secretion of protein as well. In this experiment ABC transporters are expressed from plasmids as well but proof of ABC transporter expression is not presented. Moreover a control would be interesting without ABC transporter overexpression as supplemental data.

The ABC transporters were expressed from independent plasmids, so the expression of ABC transporters and the expression of protein of interest probably did not interfere with each other. In detail, the host bacterium in Figure 8 harbored two plasmids, one harboring a gene for the protein of interest, and the other harboring the gene for the ABC transporter complex. Included is the plasmid structures:

Fig R1. A plasmid harboring genes for the ABC transporter complex and chloramphenicol resistance.

Fig R2. A plasmid used to deliver the genes for the protein of interest expression. The multiple cloning site of this plasmid was used for insertion of the genes of interest, Cuti(−) and Cuti, and the gene for LARD3 signal sequence. The Apr gene (not to be confused with the AprDEF transporter genes in Fig R1) provides the ampicillin resistance.

An indirect proof of the expression of ABC transporters is in Fig A4 in the appendix of the main text. In there, the clear halo sign of TliA lipase (which has LARD3 secretion signal) can only be seen in the colonies harboring the ABC transporters.

For the suggested comparison without the expression of ABC transporter, we believe that the “Cuti(−) only” and “Cuti only” lanes in Fig 8c-d provide a control where the ABC transporter does not exist. The proteins were not secreted if ABC transporters were not there, even if the LARD3 signal sequence was attached.

Reviewer 2 Report

The manuscript is devoted to identification of the sequence requirements for protein translocation through a bacterial membrane. The authors considered the role of positively and negatively charged residues and their abundance on the translocation. The results are very interesting and the approach proposed to regulate protein secretion may have a wide application in research in biotechnology. The study was very well planned and performed. The results are solid, and the manuscript is well written. I have only minor technical remarks.

It should be indicated which positive residues are found in the supercharged regions of the proteins in Figure 2. This is important given that Lys is less harmful than Arg.

Contrasting colors should be used in Figure 2b,c to enhance visibility.

The nonexisting term “primary sequence” should be replaced by “amino acid sequence” or “primary structure”.

Author Response

Thank you for the valuable critiques you provided. We hereby send you a point-by-point response to your review. Your original comment is colored black, while our response is colored red.

The manuscript is devoted to identification of the sequence requirements for protein translocation through a bacterial membrane. The authors considered the role of positively and negatively charged residues and their abundance on the translocation. The results are very interesting and the approach proposed to regulate protein secretion may have a wide application in research in biotechnology. The study was very well planned and performed. The results are solid, and the manuscript is well written. I have only minor technical remarks.

It should be indicated which positive residues are found in the supercharged regions of the proteins in Figure 2. This is important given that Lys is less harmful than Arg.

We added two panels as Figure 2d and 2e, describing the amino acid sequences in the highlighted regions in Figure 2b and 2c.

Contrasting colors should be used in Figure 2b,c to enhance visibility.

We changed the color scheme used in Figure 2b and 2c, enhancing the contrast.

The nonexisting term “primary sequence” should be replaced by “amino acid sequence” or “primary structure”.

We corrected the erroneous term into “amino acid sequence”.
